# SafetyNets: Verifiable Execution of Deep Neural Networks on an Untrusted Cloud

**Zahra Ghodsi, Tianyu Gu, Siddharth Garg**
New York University
{zg451, tg1553, sg175}@nyu.edu

## Abstract

Inference using deep neural networks is often outsourced to the cloud since it is a computationally demanding task. However, this raises a fundamental issue of trust. How can a client be sure that the cloud has performed inference correctly? A lazy cloud provider might use a simpler but less accurate model to reduce its own computational load, or worse, maliciously modify the inference results sent to the client. We propose SafetyNets, a framework that enables an untrusted server (the cloud) to provide a client with a short mathematical proof of the correctness of inference tasks that they perform on behalf of the client. Specifically, SafetyNets develops and implements a specialized *interactive proof* (IP) protocol for verifiable execution of a class of deep neural networks, i.e., those that can be represented as arithmetic circuits. Our empirical results on three- and four-layer deep neural networks demonstrate the run-time costs of SafetyNets for both the client and server are low. SafetyNets detects any incorrect computations of the neural network by the untrusted server with high probability, while achieving state-of-the-art accuracy on the MNIST digit recognition (99.4%) and TIMIT speech recognition tasks (75.22%).

## 1 Introduction

Recent advances in deep learning have shown that multi-layer neural networks can achieve state-of-the-art performance on a wide range of machine learning tasks. However, training and performing inference (using a trained neural network for predictions) can be computationally expensive. For this reason, several commercial vendors have begun offering "machine learning as a service" (MLaaS) solutions that allow clients to *outsource* machine learning computations, both training and inference, to the cloud.

While promising, the MLaaS model (and outsourced computing, in general) raises immediate security concerns, specifically relating to the *integrity* (or correctness) of computations performed by the cloud and the *privacy* of the client's data [16]. This paper focuses on the former, i.e., the question of integrity. Specifically, how can a client perform inference using a deep neural network on an untrusted cloud, while obtaining strong assurance that the cloud has performed inference correctly?

Indeed, there are compelling reasons for a client to be wary of a third-party cloud's computations. For one, the cloud has a financial incentive to be "lazy." A lazy cloud might use a simpler but less accurate model, for instance, a single-layer instead of a multi-layer neural network, to reduce its computational costs. Further the cloud could be compromised by malware that modifies the results sent back to the client with malicious intent. For instance, the cloud might always mis-classify a certain digit in a digit recognition task, or allow unauthorized access to certain users in a face recognition based authentication system.

The security risks posed by cloud computing have spurred theoretical advances in the area of *verifiable computing* (VC) [21]. The idea is to enable a client to *provably* (and *cheaply*) verify that an untrusted

server has performed computations correctly. To do so, the server provides to the client (in addition to the result of computation) a *mathematical* proof of the correctness of the result. The client rejects, with high probability, any incorrectly computed results (or proofs) provided by the server, while always accepting correct results (and corresponding proofs) [1]. VC techniques aim for the following desirable properties: the size of the proof should be small, the client's verification effort must be lower than performing the computation locally, and the server's effort in generating proofs should not be too high.

The advantage of proof-based VC is that it provides *unconditional*, mathematical guarantees on the integrity of computation performed by the server. Alternative solutions for verifiable execution require the client to make trust *assumptions* that are hard for the client to independently verify. Trusted platform modules [7], for instance, require the client to place trust on the hardware manufacturer, and assume that the hardware is tamper-proof. Audits based on the server's execution time [15] require precise knowledge of the server's hardware configuration and assume, for instance, that the server is not over-clocked.

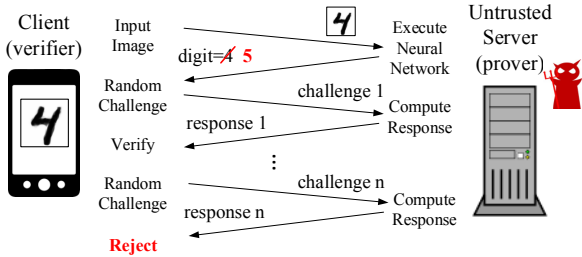

Figure 1: High-level overview of the SafetyNets IP protocol. In this example, an untrusted server intentionally changes the classification output from $4$ to $5$.

The work in this paper leverages powerful VC techniques referred to as interactive proof (IP) systems [5, 9, 18, 19]. An IP system consists of two entities, a prover ($\mathcal{P}$), i.e., the untrusted server, and a verifier ($\mathcal{V}$), i.e., the client. The framework is illustrated in Figure 1. The verifier sends the prover an input $\boldsymbol{x}$, say a batch of test images, and asks the prover to compute a function $\boldsymbol{y} = f(\boldsymbol{x})$. In our setting, $f(.)$ is a trained multi-layer neural network that is known to both the verifier and prover, and $\boldsymbol{y}$ is the neural network's classification output for each image in the batch. The prover performs the computation and sends the verifier a purported result $\boldsymbol{y'}$ (which is not equal to $\boldsymbol{y}$ if the prover cheats). The verifier and prover then engage in $n$ rounds of interaction. In each round, the verifier sends the prover a randomly picked challenge, and the prover provides a response based on the IP protocol. The verifier *accepts* that $\boldsymbol{y'}$ is indeed equal to $f(\boldsymbol{x})$ if it is satisfied with the prover's response in each round, and *rejects* otherwise.

A major criticism of IP systems (and, indeed, all existing VC techniques) when used for verifying general-purpose computations is that the prover's overheads are *large*, often orders of magnitude more than just computing $f(\boldsymbol{x})$ [21]. Recently, however, Thaler [18] showed that certain types of computations admit IP protocols with highly efficient verifiers and provers, which lays the foundations for the specialized IP protocols for deep neural networks that we develop in this paper.

**Paper Contributions.** This paper introduces SafetyNets, a new (and to the best of our knowledge, the first) approach for verifiable execution of deep neural networks on untrusted clouds. Specifically, SafetyNets composes a new, specialized IP protocol for the neural network's activation layers with Thaler's IP protocol for matrix multiplication to achieve end-to-end verifiability, dramatically reducing the bandwidth costs versus a naive solution that verifies the execution of each layer of the neural network separately.

SafetyNets applies to a certain class of neural networks that can be represented as arithmetic circuits that perform computations over finite fields (i.e., integers modulo a large prime $p$). Our implementation of SafetyNets addresses several practical challenges in this context, including the choice of the prime $p$, its relationship to accuracy of the neural network, and to the verifier and prover run-times.

Empirical evaluations on the MNIST digit recognition and TIMIT speech recognition tasks illustrate that SafetyNets enables practical, low-cost verifiable outsourcing of deep neural network execution without compromising classification accuracy. Specifically, the client's execution time is $8\times$-$80\times$ lower than executing the network locally, the server's overhead in generating proofs is less than $5\%$, and the client/server exchange less than $8$ KBytes of data during the IP protocol. SafetyNets' security

guarantees ensure that a client can detect any incorrect computations performed by a malicious server with probability vanishingly close to 1. At the same time, SafetyNets achieves state-of-the-art classification accuracies of 99.4% and 75.22% on the MNIST and TIMIT datasets, respectively.

## 2  Background

In this section, we begin by reviewing necessary background on IP systems, and then describe the restricted class of neural networks (those that can be represented as arithmetic circuits) that SafetyNets handles.

### 2.1  Interactive Proof Systems

Existing IP systems proposed in literature [5, 9, 18, 19] use, at their heart, a protocol referred to as the sum-check protocol [13] that we describe here in some detail, and then discuss its applicability in verifying general-purpose computations expressed as arithmetic circuits.

**Sum-check Protocol**  Consider a $d$-degree $n$-variate polynomial $g(x_1, x_2, \ldots, x_n)$, where each variable $x_i \in \mathbb{F}_p$ ($\mathbb{F}_p$ is the set of all natural numbers between zero and $p - 1$, for a given prime $p$) and $g : \mathbb{F}_p^n \to \mathbb{F}_p$. The prover $\mathcal{P}$ seeks to prove the following claim:

$$y = \sum_{x_1 \in \{0,1\}} \sum_{x_2 \in \{0,1\}} \cdots \sum_{x_n \in \{0,1\}} g(x_1, x_2, \ldots, x_n) \tag{1}$$

that is, the sum of $g$ evaluated at $2^n$ points is $y$. $\mathcal{P}$ and $\mathcal{V}$ now engage in a sum-check protocol to verify this claim. In the first round of the protocol, $\mathcal{P}$ sends the following unidimensional polynomial

$$h(x_1) = \sum_{x_2 \in \{0,1\}} \sum_{x_3 \in \{0,1\}} \cdots \sum_{x_n \in \{0,1\}} g(x_1, x_2, \ldots, x_n) \tag{2}$$

to $\mathcal{V}$ in the form of its coefficients. $\mathcal{V}$ checks if $h(0) + h(1) = y$. If yes, it proceeds, otherwise it rejects $\mathcal{P}$'s claim. Next, $\mathcal{V}$ picks a random value $q_1 \in \mathbb{F}_p$ and evaluates $h(q_1)$ which, based on Equation 2, yields a new claim:

$$h(q_1) = \sum_{x_2 \in \{0,1\}} \sum_{x_3 \in \{0,1\}} \cdots \sum_{x_n \in \{0,1\}} g(q_1, x_2, \ldots, x_n). \tag{3}$$

$\mathcal{V}$ now recursively calls the sum-check protocol to verify this new claim. By the final round of the sum-check protocol, $\mathcal{P}$ returns the value $g(q_1, q_2, \ldots, q_n)$ and the $\mathcal{V}$ checks if this value is correct by evaluating the polynomial by itself. If so, $\mathcal{V}$ accepts the original claim in Equation 1, otherwise it rejects the claim.

**Lemma 2.1.**  *[2] $\mathcal{V}$ rejects an incorrect claim by $\mathcal{P}$ with probability greater than $(1 - \epsilon)$ where $\epsilon = \frac{nd}{p}$ is referred to as the **soundness error**.*

**IPs for Verifying Arithmetic Circuits**  In their seminal work, Goldwasser et al. [9] demonstrated how sum-check can be used to verify the execution of arithmetic circuits using an IP protocol now referred to as GKR. An arithmetic circuit is a directed acyclic graph of computation over elements of a finite field $\mathbb{F}_p$ in which each node can perform either *addition* or *multiplication* operations (modulo $p$). While we refer the reader to [9] for further details of GKR, one important aspect of the protocol bears mention.

GKR organizes nodes of an arithmetic circuit into *layers*; starting with the circuit inputs, the outputs of one layer feed the inputs of the next. The GKR proof protocol operates backwards from the circuit outputs to its inputs. Specifically, GKR uses sum-check to reduce the prover's assertion about the circuit output into an assertion about the inputs of the output layer. This assertion is then reduced to an assertion about the inputs of the penultimate layer, and so on. The protocol continues iteratively till the verifier is left with an assertion about the circuit inputs, which it checks on its own. The layered nature of GKR's prover aligns almost perfectly with the structure of a multi-layer neural network and motivates the use of an IP system based on GKR for SafetyNets.

## 2.2 Neural Networks as Arithmetic Circuits

As mentioned before, SafetyNets applies to neural networks that can be expressed as arithmetic circuits. This requirement places the following restrictions on the neural network layers.

**Quadratic Activations** The activation functions in SafetyNets must be polynomials with integer coefficients (or, more precisely, coefficients in the field $\mathbb{F}_p$). The simplest of these is the element-wise quadratic activation function whose output is simply the square of its input. Other commonly used activation functions such as ReLU, sigmoid or softmax activations are precluded, *except* in the final output layer. Prior work has shown that neural networks with quadratic activations have the same representation power as networks with threshold activations and can be efficiently trained [6, 12].

**Sum Pooling** Pooling layers are commonly used to reduce the network size, to prevent overfitting and provide translation invariance. SafetyNets uses sum pooling, wherein the output of the pooling layer is the sum of activations in each local region. However, techniques such as max pooling [10] and stochastic pooling [22] are not supported since max and divisions operations are not easily represented as arithmetic circuits.

**Finite Field Computations** SafetyNets supports computations over elements of the field $\mathbb{F}_p$, that is, integers in the range $\{-\frac{p-1}{2}, \ldots, 0, \ldots, \frac{p-1}{2}\}$. The inputs, weights and all intermediate values computed in the network must lie in this range. Note that due to the use of quadratic activations and sum pooling, the values in the network can become quite large. In practice, we will pick large primes to support these large values. We note that this restriction applies to the inference phase *only*; the network can be trained with floating point inputs and weights. The inputs and weights are then re-scaled and quantized, as explained in Section 3.3, to finite field elements.

We note that the restrictions above are shared by a recently proposed technique, CryptoNets [8], that seeks to perform neural network based inference on *encrypted* inputs so as to guarantee data privacy. However, Cryptonets does not guarantee integrity and compared to SafetyNets, incurs high costs for both the client and server (see Section 4.3 for a comparison). Conversely, SafetyNets is targeted towards applications where integrity is critical, but does not provide privacy.

## 2.3 Mathematical Model

An $L$ layer neural network with the constraints discussed above can be modeled, without loss of generality, as follows. The input to the network is $\boldsymbol{x} \in \mathbb{F}_p^{n_0 \times b}$, where $n_0$ is the dimension of each input and $b$ is the batch size. Layer $i \in [1, L]$ has $n_i$ output neurons[2], and is specified using a weight matrix $\boldsymbol{w}_{i-1} \in \mathbb{F}_p^{n_i \times n_{i-1}}$, and biases $\boldsymbol{b}_{i-1} \in \mathbb{F}_p^{n_i}$.

The output of Layer $i \in [1, L]$, $\boldsymbol{y}_i \in \mathbb{F}_p^{n_i \times b}$ is:

$$\boldsymbol{y}_i = \sigma_{quad}(\boldsymbol{w}_{i-1}.\boldsymbol{y}_{i-1} + \boldsymbol{b}_{i-1}\mathbf{1}^T) \ \forall i \in [1, L-1]; \quad \boldsymbol{y}_L = \sigma_{out}(\boldsymbol{w}_{L-1}.\boldsymbol{y}_{L-1} + \boldsymbol{b}_{L-1}\mathbf{1}^T), \quad (4)$$

where $\sigma_{quad}(.)$ is the quadratic activation function, $\sigma_{out}(.)$ is the activation function of the output layer, and $\mathbf{1} \in \mathbb{F}_p^b$ is the vector of all ones. We will typically use softmax activations in the output layer. We will also find it convenient to introduce the variable $\boldsymbol{z}_i \in \mathbb{F}_p^{n_{i+1} \times b}$ defined as

$$\boldsymbol{z}_i = \boldsymbol{w}_i.\boldsymbol{y}_i + \boldsymbol{b}_i\mathbf{1}^T \ \forall i \in [0, L-1]. \quad (5)$$

The model captures both fully connected and convolutional layers; in the latter case the weight matrix is sparse. Further, without loss of generality, all successive linear transformations in a layer, for instance sum pooling followed by convolutions, are represented using a single weight matrix.

With this model in place, the goal of SafetyNets is to enable the client to verify that $\mathbf{y_L}$ was correctly computed by the server. We note that as in prior work [19], SafetyNets amortizes the prover and verifier costs over *batches* of inputs. If the server incorrectly computes the output corresponding to any input in a batch, the verifier rejects the entire batch of computations.

# 3 SafetyNets

We now describe the design and implementation of our end-to-end IP protocol for verifying execution of deep networks. The SafetyNets protocol is a specialized form of the IP protocols developed by Thaler [18] for verifying "regular" arithmetic circuits, that themselves specialize and refine prior work [5]. The starting point for the protocol is a polynomial representation of the network's inputs and parameters, referred to as a multilinear extension.

**Multilinear Extensions**  Consider a matrix $w \in \mathbb{F}_p^{n \times n}$. Each row and column of $w$ can be referenced using $m = \log_2(n)$ bits, and consequently one can represent $w$ as a function $W : \{0,1\}^m \times \{0,1\}^m \to \mathbb{F}_p$. That is, given Boolean vectors $t, u \in \{0,1\}^m$, the function $W(t, u)$ returns the element of $w$ at the row and column specified by Boolean vectors $t$ and $u$, respectively.

A *multi-linear extension* of $W$ is a polynomial function $\tilde{W} : \mathbb{F}_p^m \times \mathbb{F}_p^m \to \mathbb{F}_p$ that has the following two properties: (1) given vectors $t, u \in \mathbb{F}_p^m$ such that $\tilde{W}(t, u) = W(t, u)$ for all points on the unit hyper-cube, that is, for all $t, u \in \{0,1\}^m$; and (2) $\tilde{W}$ has degree 1 in each of its variables. In the remainder of this discussion, we will use $\tilde{X}, \tilde{Y}_i$ and $\tilde{Z}_i$ and $\tilde{W}_i$ to refer to multi-linear extensions of $x, y_i, z_i,$ and $w_i$, respectively, for $i \in [1, L]$. We will also assume, for clarity of exposition, that the biases, $b_i$ are zero for all layers. The supplementary draft describes how biases are incorporated. Consistent with the IP literature, the description of our protocol refers to the client as the verifier and the server as the prover.

**Protocol Overview**  The verifier seeks to check the result $y_L$ provided by the prover corresponding to input $x$. Note that $y_L$ is the output of the final activation layer which, as discussed in Section 2.2, is the only layer that does not use quadratic activations, and is hence not amenable to an IP.

Instead, in SafetyNets, the prover computes and sends $z_{L-1}$ (the *input* of the final activation layer) as a result to the verifier. $z_{L-1}$ has the same dimensions as $y_L$ and therefore this refinement has no impact on the server to client bandwidth. Furthermore, the verifier can easily compute $y_L = \sigma_{out}(z_{L-1})$ locally.

Now, the verifier needs to check whether the prover computed $z_{L-1}$ correctly. As noted by Vu et al. [19], this check can be replaced by a check on whether the multilinear extension of $z_{L-1}$ is correctly computed at a randomly picked point in the field, with minimal impact on the soundness error. That is, the verifier picks two vectors, $q_{L-1} \in \mathbb{F}_p^{log(n_L)}$ and $r_{L-1} \in \mathbb{F}_p^{log(b)}$ at *random*, evaluates $\tilde{Z}_{L-1}(q_{L-1}, r_{L-1})$, and checks whether it was correctly computed using a specialized sum-check protocol for matrix multiplication due to Thaler [18] (described in Section 3.1).

Since $z_{L-1}$ depends on $w_{L-1}$ and $y_{L-1}$, sum-check yields assertions on the values of $\tilde{W}_{L-1}(q_{L-1}, s_{L-1})$ and $\tilde{Y}_{L-1}(s_{L-1}, r_{L-1})$, where $s_{L-1} \in \mathbb{F}_p^{log(n_{L-1})}$ is another random vector picked by the verifier during sum-check.

$\tilde{W}_{L-1}(q_{L-1}, s_{L-1})$ is an assertion about the weight of the final layer. This is checked by the verifier locally since the weights are known to both the prover and verifier. Finally, the verifier uses our specialized sum-check protocol for activation layers (described in Section 3.2) to reduce the assertion on $\tilde{Y}_{L-1}(s_{L-1}, r_{L-1})$ to an assertion on $\tilde{Z}_{L-2}(q_{L-2}, s_{L-2})$. The protocol repeats till it reaches the input layer and produces an assertion on $\tilde{X}(s_0, r_0)$, the multilinear extension of the input x. The verifier checks this locally. If at any point in the protocol, the verifier's checks fail, it rejects the prover's computation. Next, we describe the sum-check protocols for matrix multiplication and activation that SafetyNets uses.

## 3.1 Sum-check for Matrix Multiplication

Since $z_i = w_i . y_i$ (recall we assumed zero biases for clarity), we can check an assertion about the multilinear extension of $z_i$ evaluated at randomly picked points $q_i$ and $r_i$ by expressing $\tilde{Z}_i(q_i, r_i)$ as [18]:

$$\tilde{Z}_i(q_i, r_i) = \sum_{j \in \{0,1\}^{\log(n_i)}} \tilde{W}_i(q_i, j) . \tilde{Y}_i(j, r_i) \tag{6}$$

Note that Equation 6 has the same form as the sum-check problem in Equation 1. Consequently the sum-check protocol described in Section 2.1 can be used to verify this assertion. At the end of the sum-check rounds, the verifier will have assertions on $\tilde{W}_i$ which it checks locally, and $\tilde{Y}_i$ which is checked using the sum-check protocol for quadratic activations described in Section 3.2.

The prover run-time for running the sum-check protocol in layer $i$ is $\mathbb{O}(n_i(n_{i-1} + b))$, the verifier's run-time is $\mathbb{O}(n_i n_{i-1})$ and the prover/verifier exchange $4 \log(n_i)$ field elements.

## 3.2 Sum-check for Quadratic Activation

In this step, we check an assertion about the output of quadratic activation layer $i$, $\tilde{Y}_i(\boldsymbol{s}_i, \boldsymbol{r}_i)$, by writing it in terms of the input of the activation layer as follows:

$$\tilde{Y}_i(\boldsymbol{s}_i, \boldsymbol{r}_i) = \sum_{j \in \{0,1\}^{\log(n_i)}, k \in \{0,1\}^{\log(b)}} \tilde{I}(\boldsymbol{s}_i, \boldsymbol{j}) \tilde{I}(\boldsymbol{r}_i, \boldsymbol{k}) \tilde{Z}_{i-1}^2(\boldsymbol{j}, \boldsymbol{k}), \tag{7}$$

where $\tilde{I}(.,.)$ is the multilinear extension of the identity matrix. Equation 7 can also be verified using the sum-check protocol, and yields an assertion about $\tilde{Z}_{i-1}$, i.e., the inputs to the activation layer. This assertion is in turn checked using the protocol described in Section 3.1.

The prover run-time for running the sum-check protocol in layer $i$ is $\mathbb{O}(bn_i)$, the verifier's run-time is $\mathbb{O}(\log(bn_i))$ and the prover/verifier exchange $5 \log(bn_i)$ field elements. This completes the theoertical description of the SafetyNets specialized IP protocol.

**Lemma 3.1.** *The SafetyNets verifier rejects incorrect computations with probability greater than* $(1 - \epsilon)$ *where* $\epsilon = \frac{3b \sum_{i=0}^{L} n_i}{p}$ *is the soundness error.*

In practice, with $p = 2^{61} - 1$ the soundness error $< \frac{1}{2^{30}}$ for practical network parameters and batch sizes.

## 3.3 Implementation

The fact that SafetyNets operates only on elements in a finite field $\mathbb{F}_p$ during inference imposes a practical challenge. That is, how do we convert floating point inputs and weights from training into field elements, and how do we select the size of the field $p$?

Let $\boldsymbol{w}_i' \in \mathbb{R}^{n_{i-1} \times n_i}$ and $\boldsymbol{b}_i' \in \mathbb{R}^{n_i}$ be the floating point parameters obtained from training for each layer $i \in [1, L]$. We convert the weights to integers by multiplying with a constant $\beta > 1$ and rounding, i.e., $\boldsymbol{w}_i = \lfloor \beta \boldsymbol{w}_i' \rceil$. We do the same for inputs with a scaling factor $\alpha$, i.e., $\boldsymbol{x} = \lfloor \alpha \boldsymbol{x}' \rceil$. Then, to ensure that all values in the network scale isotropically, we must set $\boldsymbol{b}_i = \lfloor \alpha^{2^{i-1}} \beta^{(2^{i-1}+1)} \boldsymbol{b}_i' \rceil$.

While larger $\alpha$ and $\beta$ values imply lower quantization errors, they also result in large values in the network, especially in the layers closer to the output. Similar empirical observations were made by the CryptoNets work [8]. To ensure accuracy the values in the network must lie in the range $[-\frac{p-1}{2}, \frac{p-1}{2}]$; this influences the choice of the prime $p$. On the other hand, we note that large primes increase the verifier and prover run-time because of the higher cost of performing modular additions and multiplications.

As in prior works [5, 18, 19], we restrict our choice of $p$ to Mersenne primes since they afford efficient modular arithmetic implementations, and specifically to the primes $p = 2^{61} - 1$ and $p = 2^{127} - 1$. For a given $p$, we explore and different values of $\alpha$ and $\beta$ and use the validation dataset to the pick the ones that maximize accuracy while ensuring that the values in the network lie within $[-\frac{p-1}{2}, \frac{p-1}{2}]$.

# 4 Empirical Evaluation

In this section, we present empirical evidence to support our claim that SafetyNets enables low-cost verifiable execution of deep neural networks on untrusted clouds without compromising classification accuracy.

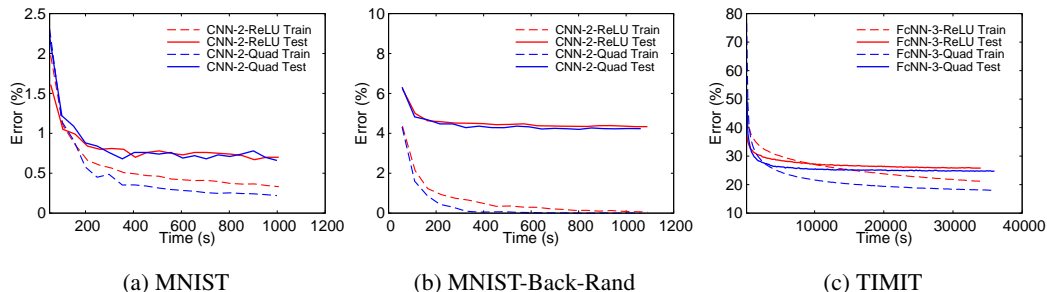

<div align="center">

(a) MNIST        (b) MNIST-Back-Rand        (c) TIMIT

</div>

Figure 2: Evolution of training and test error for the MNIST, MNIST-Back-Rand and TIMIT tasks.

## 4.1 Setup

**Datasets** We evaluated SafetyNets on three classifications tasks. (1) Handwritten digit recognition on the MNIST dataset, using 50,000 training, 10,000 validation and 10,000 test images. (2) A more challenging version of digit recognition, MNIST-Back-Rand, an artificial dataset generated by inserting a random background into MNIST image [1]. The dataset has 10,000 training, 2,000 validation and 50,000 test images. ZCA whitening is applied to the raw dataset before training and testing [4]. (3) Speech recognition on the TIMIT dataset, split into a training set with 462 speakers, a validation set with 144 speakers and a testing set with 24 speakers. The raw audio samples are pre-processed as described by [3]. Each example includes its preceding and succeeding 7 frames, resulting in a 1845-dimensional input in total. During testing, all labels are mapped to 39 classes [11] for evaluation.

**Neural Networks** For the two MNIST tasks, we used a convolutional neural network same as [23] with 2 convolutional layers with $5 \times 5$ filters, a stride of 1 and a mapcount of 16 and 32 for the first and second layer respectively. Each convolutional layer is followed by quadratic activations and $2 \times 2$ sum pooling with a stride of 2. The fully connected layer uses softmax activation. We refer to this network as **CNN-2-Quad**. For TIMIT, we use a four layer network described by [3] with 3 hidden, fully connected layers with 2000 neurons and quadratic activations. The output layer is fully connected with 183 output neurons and softmax activation. We refer to this network as **FcNN-3-Quad**. Since quadratic activations are not commonly used, we compare the performance of CNN-2-Quad and FcNN-3-Quad with baseline versions in which the quadratic activations are replaced by ReLUs. The baseline networks are **CNN-2-ReLU** and **FcNN-3-ReLU**.

The hyper-parameters for training are selected based on the validation datasets. The Adam Optimizer is used for CNNs with learning rate 0.001, exponential decay and dropout probability 0.75. The AdaGrad optimizer is used for FcNNs with a learning rate of 0.01 and dropout probability 0.5. We found that norm gradient clipping was required for training the CNN-2-Quad and FcNN-3-Quad networks, since the gradient values for quadratic activations can become large.

Our implementation of SafetyNets uses Thaler's code for the IP protocol for matrix multiplication [18] and our own implementation of the IP for quadratic activations. We use an Intel Core i7-4600U CPU running at 2.10 GHz for benchmarking.

## 4.2 Classification Accuracy of SafetyNets

SafetyNets places certain restrictions on the activation function (quadratic) and requires weights and inputs to be integers (in field $F_p$). We begin by analyzing how (and if) these restrictions impact classification accuracy/error. Figure 2 compares training and test error of CNN-2-Quad/FcNN-3-Quad versus CNN-2-ReLU/FcNN-3-ReLU. For all three tasks, the networks with quadratic activations are competitive with networks that use ReLU activations. Further, we observe that the networks with quadratic activations appear to converge faster during training, possibly because their gradients are larger despite gradient clipping.

Next, we used the scaling and rounding strategy proposed in Section 3.3 to convert weights and inputs to integers. Table 1 shows the impact of scaling factors $\alpha$ and $\beta$ on the classification error and maximum values observed in the network during inference for MNIST-Back-Rand. The validation

Table 1: Validation error and maximum value observed in the network for MNIST-Rand-Back and different values of scaling parameters, $\alpha$ and $\beta$. Shown in bold red font are values of $\alpha$ and $\beta$ that are infeasible because the maximum value exceeds that allowed by prime $p = 2^{61} - 1$.

| $\beta$ | $\alpha = 4$ | | $\alpha = 8$ | | $\alpha = 16$ | | $\alpha = 32$ | | $\alpha = 64$ | |
|---|---|---|---|---|---|---|---|---|---|---|
| | Err | Max | Err | Max | Err | Max | Err | Max | Err | Max |
| 4 | 0.188 | $4.0 \times 10^8$ | 0.073 | $4.0 \times 10^{10}$ | 0.042 | $5.5 \times 10^{12}$ | 0.039 | $6.6 \times 10^{14}$ | 0.04 | $8.8 \times 10^{16}$ |
| 8 | 0.194 | $6.1 \times 10^9$ | 0.072 | $6.9 \times 10^{11}$ | 0.039 | $8.3 \times 10^{13}$ | 0.038 | $1.0 \times 10^{16}$ | 0.037 | $\mathbf{1.3 \times 10^{18}}$ |
| 16 | 0.188 | $9.4 \times 10^{10}$ | 0.072 | $1.1 \times 10^{13}$ | 0.036 | $1.3 \times 10^{15}$ | 0.037 | $1.6 \times 10^{17}$ | 0.035 | $\mathbf{2.1 \times 10^{19}}$ |
| 32 | 0.186 | $1.5 \times 10^{12}$ | 0.073 | $1.7 \times 10^{14}$ | 0.038 | $2.1 \times 10^{16}$ | 0.037 | $\mathbf{2.6 \times 10^{18}}$ | 0.036 | $\mathbf{3.5 \times 10^{20}}$ |
| 64 | 0.185 | $2.5 \times 10^{13}$ | 0.073 | $2.8 \times 10^{15}$ | 0.038 | $3.4 \times 10^{17}$ | 0.037 | $\mathbf{4.2 \times 10^{19}}$ | 0.036 | $\mathbf{5.6 \times 10^{21}}$ |

error drops as $\alpha$ and $\beta$ are increased. On the other hand, for $p = 2^{61} - 1$, the largest value allowed is $1.35 \times 10^{18}$; this rules out $\alpha$ and $\beta$ greater than 64, as shown in the table. For MNIST-Back-Rand, we pick $\alpha = \beta = 16$ based on validation data, and obtain a test error of 4.67%. Following a similar methodology, we obtain a test error of 0.63% for MNIST ($p = 2^{61} - 1$) and 25.7% for TIMIT ($p = 2^{127} - 1$). We note that SafetyNets does not support techniques such as Maxout [10] that have demonstrated lower error on MNIST (0.45%). Ba et al. [3] report an error of 18.5% for TIMIT using an *ensemble* of nine deep neural networks, which SafetyNets might be able to support by verifying each network individually and performing ensemble averaging at the client-side.

### 4.3 Verifier and Prover Run-times

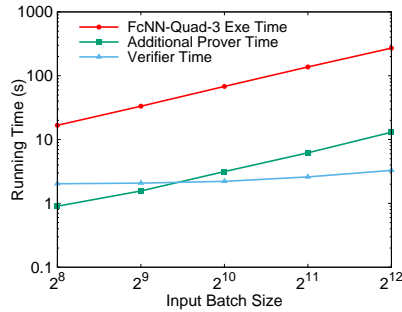

Figure 3: Run-time of verifier, prover and baseline execution time for the arithmetic circuit representation of FcNN-Quad-3 versus input batch size.

The relevant performance metrics for SafetyNets are (1) the client's (or verifier's) run-time, (2) the server's run-time which includes baseline time to execute the neural network and overhead of generating proofs, and (3) the bandwidth required by the IP protocol. Ideally, these quantities should be small, and importantly, the client's run-time should be smaller than the case in which it executes the network by itself. Figure 3 plots run-time data over input batch sizes ranging from 256 to 2048 for FcNN-Quad-3.

For FcNN-Quad-3, the client's time for verifying proofs is $8\times$ to $82\times$ faster than the baseline in which it executes FcNN-Quad-3 itself, and decreases with batch size. The increase in the server's execution time due to the overhead of generating proofs is only 5% over the baseline unverified execution of FcNN-Quad-3. The prover and verifier exchange less than 8 KBytes of data during the IP protocol for a batch size of 2048, which is negligible (less than 2%) compared to the bandwidth required to communicate inputs and outputs back and forth. In all settings, the soundness error $\epsilon$, i.e., the chance that the verifier fails to detect incorrect computations by the server is less than $\frac{1}{2^{30}}$, a negligible value. We note SafetyNets has significantly lower bandwidth costs compared to an approach that *separately* verifies the execution of each layer using only the IP protocol for matrix multiplication.

A closely related technique, CryptoNets [8], uses homomorphic encryption to provide privacy, but not integrity, for neural networks executing in the cloud. Since SafetyNets and CryptoNets target different security goals a direct comparison is not entirely meaningful. However, from the data presented in the CryptoNets paper, we note that the client's run-time for MNIST using a CNN similar to ours and an input batch size $b = 4096$ is about 600 seconds, primarily because of the high cost of encryptions. For the same batch size, the client-side run-time of SafetyNets is less than 10 seconds. Recent work has also looked at how neural networks can be trained in the cloud without compromising the user's training data [14], but the proposed techniques do not guarantee integrity. We expect that SafetyNets can be extended to address the verifiable neural network training problem as well.

## 5 Conclusion

In this paper, we have presented SafetyNets, a new framework that allows a client to provably verify the correctness of deep neural network based inference running on an untrusted clouds. Building upon the rich literature on interactive proof systems for verifying general-purpose and specialized computations, we designed and implemented a specialized IP protocol tailored for a certain class

of deep neural networks, i.e., those that can be represented as arithmetic circuits. We showed that placing these restrictions did not impact the accuracy of the networks on real-world classification tasks like digit and speech recognition, while enabling a client to verifiably outsource inference to the cloud at low-cost. For our future work, we will apply SafetyNets to deeper networks and extend it to address *both* integrity and privacy. There are VC techniques [17] that guarantee both, but typically come at higher costs. Further, building on prior work on the use of IPs to build verifiable hardware [20], we intend to deploy the SafetyNets protocol in the design of a verifiable hardware accelerator for neural network inference.

## Footnotes

[1]Note that the SafetyNets is not intended to and cannot catch any inherent mis-classifications due to the model itself, only those that result from incorrect computations of the model by the server.

[2]The $0^{th}$ layer is defined to be input layer and thus $\boldsymbol{y_0} = \boldsymbol{x}$.

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

## Proof of Lemma 3.1

**Lemma 3.1**   The SafetyNets verifier rejects incorrect computations with probability greater than $(1 - \epsilon)$ where $\epsilon = \frac{3b \sum_{i=0}^{L} n_i}{p}$ is the soundness error.

*Proof.* Verifying a multi-linear extension of the output sampled at a random point, instead of each value adds a soundness error of $\epsilon = \frac{bn_L}{p}$. Each instance of the sum-check protocol adds to the soundness error [19]. The IP protocol for matrix multiplication adds a soundness error of $\epsilon = \frac{2n_{i-1}}{p}$ in Layer $i$ [18]. Finally, the IP protocol for quadratic activations adds a soundness error of $\epsilon = \frac{3bn_i}{p}$ in Layer $i$ [18]. Summing together we get a total soundness error of $\frac{2 \sum_{i=0}^{L-1} n_i + 3 \sum_{i=1}^{L-1} bn_i + bn_L}{p}$. The final result is an upper bound on this value.                                      $\square$

## Handling Bias Variables

We assumed that the bias variables were zero, allowing us to write $bmz_i = \boldsymbol{w}_i.\boldsymbol{y}_i$ while it should be $bmz_i = \boldsymbol{w}_i.\boldsymbol{y}_i + \boldsymbol{b}_i \mathbf{1}^T$. Let $\boldsymbol{z}'_i = \boldsymbol{w}_i.\boldsymbol{y}_i$ We seek to convert an assertion on $\tilde{Z}_i(\boldsymbol{q}_i, \boldsymbol{r}_i)$ to an assertion on $\tilde{Z}'_i$. We can do so by noting that:

$$\tilde{Z}_i(\boldsymbol{q}_i, \boldsymbol{r}_i) = \sum_{j \in \{0,1\}^{\log(n_i)}} \tilde{I}(\boldsymbol{j}, \boldsymbol{q}_i)(\tilde{Z}'_i(\boldsymbol{j}, \boldsymbol{r}_i) + \tilde{B}_i(\boldsymbol{j})) \qquad (8)$$

which can be reduced to sum-check and thus yields an assertion on $\tilde{B}_i$ which the verifier checks locally and $\tilde{Z}'_i$, which is passed to the IP protocol for matrix multiplication.

