[Reviews · NeurIPS 2017]

Reviewer 1



The authors introduce a new neural network model which enables a client to verify that its request to execute inference over a neural network has been done correctly by a server. SafetyNets provide the client a high-probability guarantee of detecting wrong/lazy executions by the server, and incurs only a low overhead for the server while significantly decreasing the client's computation compared to executing the network itself. SafetyNets come with severe restrictions on the network during inference, but experimentally these restrictions are shown to have little impact on network performance. Quality: The experiments are convincing; the theory is an extension to the specific setting of neural networks of related work on interactive proofs. Clarity: This paper is very clear and easy to understand. Originality: The authors describe this as the first neural network model that provides (probabilistic) correctness guarantees. Significance: This paper appears to be an interesting contribution to the field of interactive proofs and distributed deep-learning. Detailed comments: - The experiments on MNIST and TIMIT show that using quadratic activation functions has a non-trivial, positive impact on training time and, at least for TIMIT, on test error. Have the authors evaluated their methods on more complex datasets such as CIFAR-10 (or even ImageNet?), and can they confirm that this behavior persists? - How scalable is this approach to very large (wider and/or deeper) neural networks? - Have the authors considered using regularization during training to force the weights and biases of the network to be closer to integers? It would be interesting to see whether this can further increase the neural network's final accuracy. - This is perhaps beyond the scope of this work, but providing a measure of the impact of the transformation to a SafetyNet (converting weights to integers) would be a valuable contribution: i.e., can we guarantee that this transformation only impacts the inference by at most x%, depending on the weights and biases? - SafetyNets do not provide any privacy guarantees; could they be merged with CryptoNets to provide simultaneously privacy and correctness guarantees?

Reviewer 2



This paper presents an algorithm to verify the integrity of a server that computes neural network predictions. The algorithm is based on the interactive proof scheme of Thaler. It converts the verification of a neural network to the problem of verifying the sum of polynomials, which can be carried out efficiently. The algorithm enables the user to move computation-intensive tasks to the server side, while maintaining reliability of the prediction. Although the algorithmic idea of interactive proof is not new, this paper does a good job in adapting it to the task of neural network computing, which demonstrates a nice application of the technique. The empirical results are promising, showing that reliability can be achieved with little additional computation cost and little sacrifice of accuracy. There are several limitations of the proposed algorithm. They have been discussed in the paper. The algorithm doesn't support non-polynomial activation functions, and doesn't support max pooling. However, the high-level idea is interesting enough that it may inspire a new direction of research. Hopefully these limitations can be overcome by future study. I am curious about the accuracy of baseline predictors without scaling and rounding. It is worth comparing it with accuracy of the proposed algorithm in Table 1.

Reviewer 3



The submission is interested in allowing an entity ("the verifier") to outsource the training of neural networks to a powerful but untrusted service provider ("the prover"), while obtaining a guarantee that that the prover correctly trained the neural network. Interactive proofs (IPs) are a kind of cryptographic protocol that provide such a guarantee. The dominant cost in neural network training lies in evaluating the network on many different inputs (at least, this was my understanding -- the submission could be a bit clearer about this point for the benefit of readers who are not experts on how neural networks are trained in practice). Hence, the submission is focused on giving a very efficient IP for evaluating a neural network on many different inputs. The IP given in the submission applies to a rather restricted class C of neural networks: those that exclusively use quadratic activation functions and sum-pooling gates. It does not support popular activation functions like ReLU, sigmoid, or softmax, or max-pooling or stochastic pooling. The reason for this restriction is technical: existing IP techniques are well-suited mainly to "low-degree" operations, and quadratic activation functions and sum-pooling operations are low-degree. Despite the fact that the IP applies to a rather restricted class of neural networks, the submission's experimental section states that this class of networks nonetheless achieves state-of-the-art performance on a few benchmark learning tasks. The submission represents a quite direct application of existing IP techniques (most notably, a highly efficient IP for matrix multiplication due to Thaler). The main technical contribution of the IP protocol lies in the observation that evaluating a neural network from the class C on many different inputs can be directly implemented via matrix multiplication (this observation exploits the linear nature of neurons), and squaring operations. This allows the authors to make use of the highly efficient IP for matrix multiplication due to Thaler. Without this observation, the resulting IP would be much more costly, especially for the prover, as Thaler's IP for matrix multiplication is far more efficient than general IPs for circuit-checking that have appeared in the literature. In summary, I view the main contributions of the submission as two-fold. First, the authors observe that evaluating any neural network from a certain class (described above) on many different inputs can be reduced to operations for which highly efficient IPs are already known. Second, the submission shows experimentally that although this class is restrictive, it is still powerful enough to achieve state of the art performance on some learning tasks. Evaluation: Although the submission contains a careful and thorough application of interactive proof techniques, it has a few weaknesses that I think leave it below the bar for NIPS. First, as a rather direct application of known IP methods, the technical novelty is a bit low. Second, the class of networks supported is restrictive (though, again, the experiments demonstrate that the class performs quite well on some learning tasks). Third, and perhaps most importantly, is an issue of motivation. The interactive proof in the submission allows the verifier to ensure that the neural network was correctly trained on a specific training set using a specific training algorithm. But people shouldn't care *how* the network was trained -- they should only care that the trained network makes accurate predictions (i.e., has low generalization error). Hence, there is an obvious alternative solution to the real problem that the submission does not mention or compare against: the prover can simply send the final trained-up network to the verifier. The verifier can check that this final network makes accurate predictions simply by evaluating it on a relatively small randomly chosen selection of labeled examples, and confirming that its error on these labeled examples is low. Based on the reported numbers for the verifier (i.e., an 8x-80x speedup for the verifier compared to doing the full training itself with no prover) this naive solution is in many cases likely to be more efficient for the verifier than the actual solution described in the paper.